# Assistive Systems for Visually Impaired Persons: Challenges and Opportunities for Navigation Assistance

**DOI:** 10.3390/s24113572

**Published:** 2024-06-01

**Authors:** Gabriel Iluebe Okolo, Turke Althobaiti, Naeem Ramzan

**Affiliations:** 1School of Computing, Engineering and Physical Sciences, University of the West of Scotland, Paisley PA1 2BE, UK; naeem.ramzan@uws.ac.uk; 2Department of Computer Science, Northern Border University, Arar 91431, Saudi Arabia; turke.althobaiti@nbu.edu.sa

**Keywords:** visually impaired person, artificial intelligence, Internet of Things (IoT)

## Abstract

The inability to see makes moving around very difficult for visually impaired persons. Due to their limited movement, they also struggle to protect themselves against moving and non-moving objects. Given the substantial rise in the population of those with vision impairments in recent years, there has been an increasing amount of research devoted to the development of assistive technologies. This review paper highlights the state-of-the-art assistive technology, tools, and systems for improving the daily lives of visually impaired people. Multi-modal mobility assistance solutions are also evaluated for both indoor and outdoor environments. Lastly, an analysis of several approaches is also provided, along with recommendations for the future.

## 1. Introduction

At least 2.2 billion individuals worldwide suffer from vision impairment, according to the World Health Organisation (WHO). Some of the conditions that can cause vision impairment and blindness include diabetic retinopathy, cataracts, age-related macular degeneration, glaucoma, and refractive errors [1]. The quality of life of persons with visual impairment is greatly affected, including their capacity for employment and interpersonal interactions [2]. There is an urgent need for efficient assistance for visually impaired people, due to the rapid rise in their population. For their everyday activities, the majority of visually impaired persons use some form of assistive technology [3].

For those who have low eyesight, a navigation system is essential, since it may help them avoid obstructions and offer them exact directions to their destination. A difficult problem is the creation of navigation systems that allow the visually impaired to be guided through indoor and/or outdoor environments, so they may move in unfamiliar surroundings. Scientists are still working to create a system that would enable visually impaired individuals to be more independent and aware of their environment [4].

Technological advancements in information technology (IT), particularly in mobile technology, are expanding the potential of IT-based assistive solutions to enhance the quality of life for visually impaired people. Technology can increase people’s ability to live freely and fully participate in society [5]. The scope and variety of help that IT-based assistive technologies can offer is extensive. The paradigm of humanistic computer interaction where technology blends into society has grown in popularity [6].

In addition to producing significant advances, various studies conducted over the past few years have improved the categories of assistive devices that have been produced and are currently in widespread use. To facilitate travel, assistive technologies are often categorized into three categories: electronic orientation aids, position locators, and electronic travel aids devices. According to Elmannai et al. [7], these categories include devices that collect and transmit environmental information to the user, devices that give directions to pedestrians in unfamiliar areas, and devices that pinpoint the exact location of the holder, respectively. A distinct classification of assistive technologies was put forth by Tapu et al. [8] of perceptual tools that replace vision with other inputs from the senses, and conceptual tools that create orientation strategies to represent the environment and prepare for unforeseen circumstances during navigation.

To assist those who are visually impaired, assistive devices require three components. The first is the navigation module, also known as wayfinding, which was described by Kandalan et al. [9] as a sequence of effective movements needed to reach a desired location. It makes use of the user’s initial location and is always updated. The component should ensure safe navigation and completion within a reasonable time frame. It should be able to operate both inside and outdoors, in different lighting conditions (day/night, rain/sun, etc.), regardless of whether the place has been visited before, and conduct real-time analysis without compromising resilience and accuracy.

Detecting objects is the second component. In the first place, this makes it possible to avoid hazards and provides a helpful warning to the user. It has to be able to identify both stationary and moving obstacles, as well as their positions (ideally up ahead), characteristics, and distances, to give prompt feedback. To help visually impaired people develop cognitive maps of a location and get a thorough grasp of their surroundings, object detection should also offer scene descriptions upon user request.

The final elements are the user-controlled tools and the human–machine interface. These are made up of many devices designed to gather data, process the data, and then provide the results back to the user. Various combinations may be made, and they must be selected based on several factors, including the wearability of the devices, user choice, and computational approaches.

The idea of assistive technology is used to provide support to people with impairments. With the use of mobile assistive systems, visually impaired people can take advantage of the discreet, lightweight, and portable assistance that is provided through widely used gadgets.

A comprehensive assessment of recent developments in indoor navigation was carried out by Khan et al. [10], who covered state-of-the-art techniques, difficulties, and potential directions. They classified existing research, evaluated article quality, and identified key issues and solutions. Messaoudi et al. [11] reviewed various technologies and tools for the visually impaired, focusing on advancements in mobility aids. They explored location methods, obstacle recognition, and feedback mechanisms, alongside discussing specific tools like smart canes and voice-operated devices. Isazade et al. [12] provided an extensive review of the current navigation technologies designed to assist visually impaired individuals. They covered a range of solutions, including mobile applications, web services, and various assistive devices, highlighting the importance of these technologies in enhancing accessibility.Tripathi et al. [13] carried out a systematic review of various technologies aimed at aiding the visually impaired in navigating obstacles. They evaluated studies by categorizing technologies into vision-based, non-vision-based, and hybrid systems. The review identified gaps in current research and suggested improvements for future navigation aids, emphasizing the importance of integrating advanced technologies, to provide visually impaired people with safer and more effective navigation options. According to Soliman et al. [14], assistive technology for the visually impaired has seen recent advancements. They reviewed various systems, focusing on their purposes, hardware components, and algorithms, and categorized them into environment identification and navigation assistance systems. A thorough review was carried out by Zahabi et al. [15], in order to define criteria for creating navigation applications that are specifically suited for people with impairments. It looked at both research projects and applications that are sold commercially, in a methodical manner.

The review papers highlighted above are not without their shortcomings, which are as follows: Some of them lacked a comparative analysis of user experience, long-term effectiveness, efficiency, scalability, reliability, cost-effectiveness, accessibility of the systems, and integration of these technologies into daily life. We also observed an oversight in examining the limitations and drawbacks of systems. Comparative analysis with existing systems was not covered in some of the reviews, without identifying potential areas for improvement.

Given the rapid advancement in technology and changing societal needs, we thought it necessary to provide another review paper, as this can address gaps in the current literature by incorporating the latest developments in assistive technologies, integration, and user-centric design approaches. We examined how well these technologies work and how accessible they are in actual situations, taking into account the various demands of those with disabilities. Researchers and developers who are looking to improve navigation aids and raise the standard of living of people with impairments would greatly benefit from such a review.

This research was motivated by the desire to enhance the mobility and independence of visually impaired individuals in navigating various settings. New solutions that provide potential advantages as electronic travel aids and assistive gadgets emerge as technology advances. We sought out weaknesses in existing systems and difficulties with navigation and integrating these tools and technologies. This paper can drive future developments by assessing current technologies and user experiences, ensuring that new systems not only make use of state-of-the-art technology but are also customized to satisfy the practical needs of people who are visually impaired. To help those who are visually impaired, we provide a comprehensive overview of the most recent developments and research in the areas of mobile technology, Internet of Things (IoT) devices, and artificial intelligence (computer vision).

In this paper, we review the most recent articles in navigation systems for visually impaired people. To do this, we looked at the systems that have been developed recently, taking into account several tools and technologies. We also highlight the limitations and efficiency of these systems. This work’s contribution is summed up as follows: (i) comparing different technologies and assistive tools for visually impaired people; (ii) analysis of artificial-intelligence-based object detection systems for the visually impaired; (iii) analysis of Internet of Things (IoT)-based navigation systems for visually impaired persons.

The organization of this paper is as follows: Section 2 discusses assistive tools, technologies, and systems for the visually impaired. This section also classifies assistive systems based on their technologies. Section 5 proceeds with discussions and recommendations. Section 6 gives the Conclusions.

## 2. Assistive Technologies for Visually Impaired Persons

Navigation technologies for the visually impaired can improve their users’ mobility, spatial awareness, give real-time information about their surroundings, identify impediments, and recommend the best paths. These systems incorporate several different components, including sensors, algorithms, and feedback mechanisms. To make up for the absence of visual clues, this all-encompassing strategy combines hardware and software solutions.

### 2.1. Sensing Inputs

Visually challenged people frequently employ assistive tools that use depth cameras, Bluetooth beacons, radio frequency identification (RFID), ultrasonic sensors, infrared sensors, general cameras (or mobile phone cameras), and so on.

#### 2.1.1. Smartphones

Smartphones are capable of producing high-quality, compact photos with good resolution. However, the limitation of a standard smartphone camera is that it does not have depth information, which prevents these sorts of algorithms from determining an object’s distance from the user. Generally, when navigating, general camera views are processed to identify only the obstructions in front.

#### 2.1.2. Radio Frequency Identification (RFID)

RFID is a technique in which smart labels or tags with digital data embedded in them are detected by a reader using radio waves. This technology has issues with slow read rates, collision tags, changing signal accuracy, interruptions of communications, etc. Furthermore, in the context of navigation, the user must be informed of the position of the RFID reader [16].

#### 2.1.3. Depth Camera

A depth camera offers a variety of data. Microsoft Kinect sensor [17] is frequently utilized as the main recognition hardware in depth-based vision analysis systems, among depth camera recognition systems. When it comes to obstructions, depth photos can provide more details than just two-dimensional photographs. The inability of Kinect cameras to function in brightly lit areas is one of their primary drawbacks. Light detection and ranging (LiDaR) cameras are another type of depth-based camera used for depth analysis, in addition to Kinect cameras. Both Kinect and LiDaR-based devices have the drawback of being very large, making them difficult to operate and requiring a person to navigate.

#### 2.1.4. Infrared (IR) Sensor

An IR sensor may either detect or produce an IR signal, to sense certain properties of its environment. IR-based gadgets have the disadvantage of being susceptible to interference from both artificial and natural light. Because many tags need to be installed and maintained, IR-based systems are expensive to install [18]. In addition to detecting motion, infrared sensors are capable of measuring the amount of heat an object emits.

#### 2.1.5. Ultrasonic Sensor

Ultrasonic sensors determine obstacle distance and provide the user with an audible or vibrational signal that indicates an object is ahead. This technology uses the concept of high-frequency reflection to identify impediments. To make navigating easier, certain suggestions or instructions are provided in vibrotactile form. One drawback of these systems is that, because of the ultrasonic beam’s broad beam angle, they are unable to identify obstructions with precision. In addition, these systems are unable to distinguish between different kinds of barriers, such as a bicycle or a car [19].

#### 2.1.6. RGB-D Sensor

These navigation systems are built on a range-extending RGB-D sensor. Range-based floor segmentation is supported by consumer RGB-D cameras to gather range data. In addition, the RGB sensor can identify colors and objects. Sound map data and audio directions or instructions are used to provide the user interface [20].

### 2.2. Feedback

Obstacles can be identified by a sensing input that possesses vision and non-visual capabilities during navigation. The users need to obtain notifications, along with direction signals. Three common forms of feedback include vibrations, sounds, and touch. Certain systems employ a blend of these, providing the user with a multi-modal choice for obtaining navigation signals.

#### 2.2.1. Audio Feedback

Speakers or earbuds are typically used by navigation systems to provide audio feedback. Audio feedback can be distracting for the user if they are overloaded with information, and it can also be unpleasant if they are oblivious to background noise because of aural signals [21]. To some extent, certain noise-reduction headphones are used by several navigation systems to mitigate this issue, so that the audio stream can be heard by the user without interference [22].

#### 2.2.2. Tactile Feedback

Certain navigation feedback systems also employ tactile feedback, which is provided by the user’s fingers, hand, arm, foot, or any other part of their body where pressure can be felt. Sensing feelings at different body pressure locations enables the user to avoid obstacles and receive notifications for navigational cues. In contrast to audio feedback techniques, tactile feedback can be employed to prevent user attention being distracted by ambient noises [23].

#### 2.2.3. Vibration Feedback

Given the widespread usage of smartphones in navigation system design, various systems have attempted to provide users with navigation feedback by utilizing a vibration function [24]. Certain directional indications can be provided by the vibration pattern.

#### 2.2.4. Haptic Feedback

WeWalk, a commercial device, uses an ultrasonic sensor to assist the visually impaired in detecting impediments above chest level [25]. With the aid of a mobile app, this device can also be paired via Bluetooth with a cane. The haptic feedback used in communicating with the user is vibration, to help them become aware of their surroundings. In order to alert the VI user about their surroundings, SmartCane [26], a different commercially available device, employs haptic feedback and vibration, in addition to light and ultrasonic sensors. The author of [27] described a technique that helps visually impaired people utilize haptic inputs.

### 2.3. RFID-Based Map-Reading

RFID helps people with vision impairments participate in both indoor and outdoor activities. This device uses inexpensive, energy-efficient sensors to assist the visually impaired in navigating their environment. This RFID technology cannot be used in the landscape spatial region, because of its low communication range. RFID technology [28] helps impaired individuals by providing and facilitating self-navigation in indoor environments. This method was developed to manage and handle internal navigation issues, while accounting for the dynamics and accuracy of different settings.

### 2.4. Communication Networks

Ultra-wideband (U.W.B.) sensors, Bluetooth, cellular communication networks, and Wi-Fi networks are some of the wireless-network-based methods used for indoor location and navigation [29]. When using a wireless network, indoor location is quite user-friendly for visually impaired individuals. Mobile phones may connect with one another using a cellular network system [30]. One research paper [31] claimed that using Cell-ID, which functions in the majority of cellular networks, is an easy technique for localizing cellular devices. The research in [32] suggested a hybrid strategy to enhance indoor navigation and positioning performance. This strategy combines Bluetooth, wireless local area networks, and cellular communication networks. However, because of radio frequency signal range and cellular towers, such placement is unreliable and has a large navigational error.

## 3. Assistive Tools for the Visually Impaired

According to [33], there are several drawbacks related to user participation and program adaptation. The difficulties include mobility, comfort, adaptability, and multiple feedback options. The population of visually impaired people may not have accepted the technology, and those who were supposed to utilize it may not have taken to it for various reasons.

### 3.1. Braille Signs

Directions are difficult for people with visual impairments, thus they must be remembered. Navigation assistance is the primary course of action. These days, Braille signage is installed in many public areas, including emergency rooms, train stations, educational facilities, doors, elevators, and other parts of the infrastructure, to make the navigation easier for those with visual impairments.

### 3.2. Smart Cane

With the use of smart canes, people who are visually impaired can better navigate their environment and recognize objects in front of them, regardless of size, something that is difficult to do with traditional walking sticks. When an impediment is detected by an intelligent guiding cane, the microphone in the cane’s intelligent system emits a sound. The cane aids in determining whether an area is bright or dark [34]. An innovative cane navigation system that makes use of cloud networks and IoT was presented [35] for indoor use. The cloud network is connected to an IoT scanner, and the intelligent cane navigation system is able to gather and transfer data to it.

### 3.3. Images into Sound

Depth sensors produce images that people typically see and handle. Since sound can accurately direct people, several designs translate spatial data into sound. A method to enhance navigation in the absence of vision was presented by Rehri et al. [36]. A straightforward yet effective technique for image identification using sound was described in Nair et al. [37] for visually impaired people, enabling them to see with their ears.

### 3.4. Blind Audio Guidance

A blind audio guidance system uses an ultrasonic sensor to measure distance, an automatic voice recognition sound system to offer aural instructions, and an infrared receiver sensor to detect objects. This is based on an embedded system. The optical signals are first detected by the ultrasonic sensors, which translate them into audio data.

### 3.5. Voice and Vibration

Due to their heightened sensitivity to sound, those who are visually impaired receive alerts from this type of navigation system through vibrational and vocal inputs, both indoors and outdoors. Users can move about in an alert manner and choose from various levels of intensity [38].

Figure 1 Highlighted assistive navigational components for visually impaired persons.

## 4. Assistive Systems Useful for Navigation for Visually Impaired Persons

Individuals with visual impairments have challenges when commuting, such as identifying items and navigating around barriers, including uneven surfaces, crosswalks, and objects on the surface, even with a cane. Technological advancements such as the Internet of Things and artificial intelligence can improve accessibility issues for the disabled and visually impaired. These solutions include smartphone applications, object-recognition techniques, and Internet of Things (IoT)-enabled smart cane systems.

### 4.1. Artificial Intelligence-Based Object Detection Systems for Visually Impaired Persons

Assistive technology powered by artificial intelligence can identify items, pedestrians, signs, and scenes. Deep CNN models recognize items belonging to several classes.

Tapu et al.’s DEEP-SEE framework [39] identified moving and static objects using YOLO (you look only once) for outdoor navigation. Saleh et al. [40] proposed a navigational path detection system for the visually impaired using fully convolutional networks. A technology for automated assistance was proposed by Joshi et al. [41] to identify items in real time. A deep learning model containing several object images was employed. They employed object identification in addition to a distance-measuring sensor. They recorded an accuracy of 95.19% for object detection.

A visually impaired navigation system using YOLO, an ultrasonic sensor, and Raspberry Pi was presented by R. Parvadhavardhni et al. [42]. By combining these technologies, the system becomes more accurate and efficient, enabling visually impaired individuals to navigate their environment. With the use of the information supplied by the segmented path, visually impaired people were able to locate possible walking routes. The fundamental properties of this system precluded the possibility of obstacle identification. The method was not suitable where there was bright ambient light, even if the object could be recognized because of the distortion of its organized light source, Cornacchia et al. [43].

Jiang et al. [44] processed stereo-analyzed pictures from binocular sensors through the use of a cloud-based system. The end user received warning messages about 10 distinct types of obstacles on their smartphone thanks to the CNN-based object detection cloud services.

An approach with an individual stick utilizing a Raspberry Pi, camera, Arduino, and ultrasonic sensor was proposed by Masud et al. [45]. They made use of Voila Jones and TensorFlow object detection. They employed an ultrasonic sensor to determine an object’s distance from the user and a buzzer to produce sound. It beeps twice if there is an obstruction to the user’s left, and three times if there is one to their right. They reported a 91% average performance in object detection using COCO datasets. A CNN-based visual localization framework for the visually impaired was developed by Lin et al. [46]. After several tests, GoogleNet proved to be the most accurate and user-friendly visually impaired person finder.

Srikanteswara et al. [47] proposed a model to address the challenges experienced by the visually impaired. They recorded footage using a camera, and then they transmitted the pictures of the video frames to identify data, and the user received voice feedback. They used the YOLO framework, and OpenCV. They found that a straightforward interface may be used by anyone, and that visually impaired people could identify objects in video frames with 90% accuracy. Alhichri et al. [48] proposed a method that makes use of inertial measurement sensors, a laser rangefinder, and a wide-angle camera to detect obstacles within buildings. Using the VGG16 and SqueezeNet models, it implemented ordered weighted averaging in the output stage. Overall, it performed well, correctly identifying barriers with an accuracy of 80.69%. A living-based indoor navigation system for the visually impaired that includes item extraction, object identification, a braille conversation algorithm, and output generation was proposed by Lee et al. [49]. They found that the average accuracy was 85%, the average duration for braille discussion was 6.6 s, and the accuracy of object identification was 90% or higher.

Neha et al. [50] suggested a method allowing visually impaired people to identify barriers in front of them by utilizing a smartphone. It recognizes safe zones and pedestrian lanes and gives the user sensory feedback. Using learned objects from the OpenCV library, this navigation system detects things and assists the user in identifying real-time video. Upon testing the model, they found that 83% of visually impaired users found it beneficial. A smartphone-based outdoor navigation system was proposed by Chen et al. [51] to assist visually impaired individuals in avoiding potentially harmful objects. To detect objects, they employed the SSD-MobileNetV2 dataset, which consists of 4500 images, to obtain photographs of obstacles such as cars, motorcycles, pedestrians, and electric vehicles. An 84.6% MAP (mean average precision) was noted during the model testing. They noticed that certain things were missing, since the image had fewer characteristics and tiny objects occupied the few available pixels.

A guiding system utilizing image-to-speech algorithms in smartphone cameras was proposed by Denizgez et al. [52], where audio commands are sent to the user based on their proximity to the target item. The navigation system directs the user toward the desired objects by identifying objects that are inside the camera’s field of vision and angle on a smartphone. Using techniques from image processing and deep learning, the positions of objects are determined. A CNN was utilized to detect objects for the system. They created a model based on the SSD technique, as, when it comes to object recognition, SSD offers a greater frame rate per second than Fast R-CNN. Using TensorFlow-Lite, they trained the model using the COCO dataset, which has 91 object classes. A technique was presented by See et al. [53] to assist visually impaired individuals in determining the location of obstacles within 1.6 m. Gestures and voice commands were used to operate the system. Salunkhe et al. [54] proposed an Android application that helps the visually impaired comprehend their surroundings. The application uses the TensorFlow API for object detection to identify items using a smartphone camera. The user can hear the items that the smartphone has detected through an audio output. They noted a 90% accuracy rate and 74.3% mAP (mean average precision). They created mobile-compatible object identification using TensorFlow Lite and found the model to be user-friendly.

Table 1 summarizes navigation systems for people with vision impairments based on artificial intelligence.

### 4.2. IoT-Based Navigation Systems for Visually Impaired Persons

The visually impaired can now navigate more easily with the help of Internet-of-Things-based applications and systems. BlinDar, which was developed by Saquib et al. [55], is one such application. By utilizing an Arduino IDE and ultrasonic sensors coupled with a Wi-Fi module, they developed a smart stick utilizing Internet of Things technologies. After seeing how the sensors functioned, they programmed the microcontroller to play the phrase “Obstacle Ahead” whenever an obstacle crossed a 100 cm barrier. When immediate help is needed, the stick can also sound an alarm. Kunta et al. [56] made use of an RF remote control, an Arduino microcontroller, an IR sensor, a speaker, a push button to notify emergency contacts, a vibration motor, a buzzer, a GSM module to track the location, and a soil moisture sensor. They developed a simple assistive smartphone application that lets users choose which alarm messages to send and to update contact details using a unique stick ID.

Chava et al. [57] developed a smart shoe system. Microcontrollers equipped with sensors and buzzers were utilized. The smart shoe uses buzzers to alert visually impaired users when an obstacle is spotted They used integrated sensors and the Internet of Things to create smart eyewear that increased the effectiveness of the smart shoe. To make sure the user stays clear of obstacles, these smart shoes and smart glasses interact with one another.

Kumar et al. [58] proposed an Internet of Things (IoT) wearable navigation system for visually impaired people that uses an NEO-6M GPS module to determine the user’s current location in real time. They noted a 90% accuracy rate. They made use of Raspberry Pi’s built-in text-to-speech feature for audio output, as well as an ultrasonic sensor and camera. Using a Raspberry Pi camera, they could recognize objects using TensorFlow and deep learning techniques. In the future, they want to use this technique for walking sticks.

Rahman et al. [59] proposed the use of IoT and deep learning techniques to link a smart cap with a smart cane. The smart cap used Bluetooth, Wi-Fi, and deep learning techniques. They attained a system usability scale score of 86%.

Krishnan et al. [60] presented an Internet of Things (IoT)-based navigation system that uses an Arduino, vibration sensor, piezo buzzer, and push button to notify the user’s exact position to their care home, for urgent emergency assistance. This system recognizes surface barriers in the immediate vicinity. Using voice instructions and a buzzer, the system sends out an alarm. GPS is used to determine the exact position.

Using a smart stick, Pathak et al. [61] proposed a method to identify stairs and obstacles close to the user. The stick uses infrared and ultrasonic sensors. They suggested a cheap, responsive navigation system with low power usage. Alarm feeds were utilized to notify users if an obstruction is detected. They employed an Arduino microcontroller and ultrasonic sensors to identify impediments.

A theoretical framework for navigation with electronic assistance was presented by Mala et al. [62]. They used Bluetooth connected to a GPS-integrated stick and headphones. The model facilitates the visually impaired in navigating to where they are going. Ultrasonic sensors, a GPS module for navigation, a speech synthesizer, a GPS receiver, and a microcontroller for the stick were all utilized. GPS determines the user’s location, and they can receive audio instructions to lead them to their destination via a headset.

An Internet-of-Things-based solution with wearable glasses and walking assistance for visually impaired people was proposed by Zhou et al. [63]. The AVR chip in the glasses is used to link the sensors. A portable GPS module is used to collect data from the glasses and walking stick, and a GPRS module is used to obtain position information via the cloud. The user receives voice commands to direct them using their smartphone’s portable GPS module.

Rahman et al. [64] designed a stick for the visually impaired. It uses an Arduino NANO microcontroller, sensors, and GSM to determine the user’s location and relay information. Additionally, a light sensor was included in the stick so that others may assist the user by flashing an LED. They recommended the model since it is easily available, cost-effective, and had good object detection.

Chaurasia et al. [65] proposed an indoor autonomous navigation system. Three modules were created for the system: an RFID-based destination detection module; an object identification module utilizing ultrasonic sensors; and a navigation module using a Raspberry Pi, ultrasonic sensor, and camera. When an item is detected, the model uses offline embedded text-to-speech technology to give orders to the user, indicating which way to turn.

A creative solution for visually impaired people was developed by Messaoudi et al. [66], who gave them a smart white cane and sound buzzers, to allow them to interact with their surroundings. They made use of accelerometers, cameras, and microcontrollers. They reported that the gadget alerts the user when an item is spotted by means of buzzers.

A cost-effective and user-friendly assistance system using GPS, vibration motors, buzzers, and ultrasonic sensors was presented by Saud et al. [67] for those who are visually impaired. They used the vibrating motors on a walking stick and ultrasonic sensors attached to a hat. The vibration motors vibrate in response to the detection of an obstruction. To direct the user, they utilized two switches on the walking stick to transmit signals to the phone. To guide the user, they created an application with MIT App Inventor.

Table 2 provides an overview of IoT-based navigation systems for those with visual impairments.

An ultrasonic sensor was observed for use with other IoT devices. This was due to the benefits and safety functions it possessed, which are as follows: environmentally resilient, materially independent, and non-contact measurement [68].

## 5. Discussions and Recommendations

### 5.1. Discussions

Even though there has been a lot of research performed on navigational systems designed to help visually impaired people, not all of them are in use. The majority of the strategies make sense in principle; in practice, they could be too hard or time-consuming for the user to utilize. Sections about sensing techniques, object recognition, physical hardware, and cost-effectiveness were all included in this assessment.

It was observed many systems opted for feedback methods that used audio. Some actually made use of multi- or dual-mode feedback techniques. A superior navigation system may be identified, among other things, by selecting the right feedback mechanism. However, we also should not limit people to just one way of providing feedback. In certain circumstances, one feedback technique may prevail or be more advantageous than another. For instance, an auditory feedback approach is not a good fit in a loud metropolitan setting.

The availability of assistive technology for the visually impaired is highlighted by this study, as well as their capacities and security. Assistive devices such as smart canes and other devices for the visually impaired can help them become more independent. Deep learning methods such as YOLO, SSD, and FasterR-CNN have enabled real-time detection and object recognition, and location determination can be integrated with built-in GPS, hence increasing the efficiency and potential of these technologies.

### 5.2. Recommendations

To arrive at a location safely is the primary goal of navigation. During navigation, it may be desirable for visually impaired persons to be notified about any changes in their surroundings, such as traffic jams, barriers, warning situations, etc. During navigation, users should be given the appropriate quantity of information at the appropriate time regarding their surroundings [69].

A system that only offers one type of feedback may not be helpful in many situations. While some persons rely on tactile or vibratory modes, others rely on auditory modes. However, using multiple feedback modes may become significant depending on the circumstances. Thus, if numerous feedback modalities are available, the user will be free to select one depending on the circumstances or surroundings. As a result, the system will function better in a variety of settings. Users would benefit from a multi-modal feedback system implementation [70].

When designing a navigation system for the visually impaired, one of the most crucial user-centric considerations is the time required for system familiarization. Visually impaired people need to master use of most of the navigation-assistive technologies on the market today. One of the factors that must be taken into account in this situation is that customers should not find it difficult to learn a new system, and it should not be too complicated.

A lot of the current systems are bulky or cumbersome. Another key element influencing the usefulness and user adoption of these devices is portability. Even though a lot of systems are mobile phone-linked, using them may not be as simple as one may think. A system should be affordable and made portable for easy use [71]. It should not feel as though the user is carrying a heavy burden, since this might cause great discomfort.

All things considered, the development of assistive devices for the visually impaired is an important field of study, and with ongoing technological progress, millions of individuals might see an improvement in their standard of living.

## 6. Conclusions

This paper presents a number of the most recent assistive technologies for the visually impaired in the areas of artificial intelligence (computer vision), IoT devices, and mobile devices. The most recent developments in navigation systems, tools, and technologies have also been discussed here.

Our research evaluation, which carefully evaluated the indoor and outdoor assistive navigation techniques employed by visually impaired users, analyzed numerous tools and technologies helpful for visually impaired persons. A review of other studies published on the same subject was provided. We looked at research’s limitations and drawbacks, datasets, and various methodologies. This review demonstrates that this field of study is having a significant influence on both the research community and society at large.

## Figures and Tables

**Figure 1 sensors-24-03572-f001:**
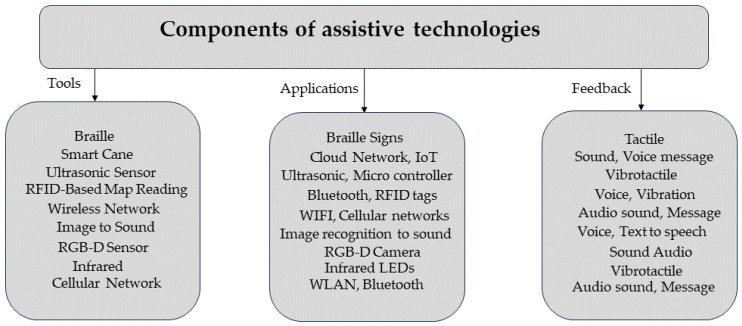
Tools and technologies for visually impaired persons.

**Table 1 sensors-24-03572-t001:** Summary of artificial intelligence-based obstacle detection systems for visually impaired persons.

Reference	Approach/Technology	Dataset	Results	Drawbacks/Limitations
[39]	DEEP-SEE framework	YOLO based object	90.00% accuracy	No early warning and requires high computational resources
[40]	Segmentation Technique	SUN RGB-D dataset	91.00% accuracy	Unsuitable for identifying far-off obstructions
[41]	YOLO-based object detection	Imagenet and COCO	95.19% accuracy	Limited object detection
[43]	Structured light source	Custom videos	95.97% accuracy	Unsuitable for strong light detection and limited data
[44]	Binocular Vision sensors	Dataset of 10 classes	67.50% precision	Restricted to 10 classes and limited by a short range
[45]	TensorFlow Object Detection	Coco dataset	91% accuracy	Long processing time
[46]	CNN-based visual localization	Imagenet Dataset	57.20% accuracy	does not detect obstacles
[47]	YOLO, OpenCV, pyttsx3	YOLO based object	90% accuracy	Distance of object no calculated
[48]	Inertial measurement sensors, CNNs	KSU and UTrentol	80.69% accuracy	The distance of the object is not calculated and no navigational aid
[49]	Braille conversation algorithm/	MS-COCO datasets	85% accuracy	Limited images for identification
[50]	obstacle and safe zone detection	OpenCV dataset	83% accuracy	Distance of object not calculated
[51]	Outdoor navigation to avoid obstacles, SSD-MobileNetV2	4500 images	84.6% mAP	Limited obstacle detection and Distance of object not calculated
[52]	Image to speech navigation	COCO, Imagenet, Pascal VOC dataset	71.5% accuracy	Distance of object not calculated
[53]	Obstacle position detection using gestures and voice	Custom COCO SSD MobileNet v2	80% accuracy	Limited to locating obstacle only
[54]	Object detection app using TensorFlow, SSD algorithm	COCO dataset	74.3% mAP	Distance of object not calculated

**Table 2 sensors-24-03572-t002:** Summary of IoT-based navigation systems for visually impaired persons.

Reference	Approach/Technology	Sensors and Libraries Used	Results	Drawbacks/Limitations
[55]	Smart stick obstacle detection	Ultrasonic sensors, Arduino IDE	Obstacle detection and sends alert	Device’s effectiveness in noisy environments and battery life
[56]	RF remote control, to measure the object’s distance from the user	Ultrasonic sensor, vibration motor, GSM module, IR sensor, Soil moisture	Sends alert messages through SMS	GPS module sometimes takes 30 s to 1 min to acquire a satellite lock
[57]	Smart shoe and smart glasses	Ultrasonic sensors and Arduino	Smart shoe warns user using buzzers when obstacle detected	Object detection and navigation guidance not supported
[58]	Wearable navigation system	NEO-6M GPS module, ultrasonic, Raspberry Pi camera, TensorFlow	90% accuracy observed	Detecting moving objects inaccurately when measured at an angle. Computational cost due to neural network.
[59]	Smart cap with smart cane	GPS module, ultrasonic sensor, Raspberry Pi camera	86% System Usability scale score	limited real-world images, for object detection
[60]	Monitoring system	Arduino, Vibration sensor, piezo buzzer, ultrasonic sensor	Notify the user’s exact position to their care home.	Distance of objects not calculated
[61]	Smart stick to detect obstacles	Ultrasonic sensors, Arduino, Infrared sensor	low cost and good navigation	Does not provide navigational guidance, and the range of sensors and the types of obstacles it can detect are limited.
[62]	GPS embedded stick	GPS module, speech synthesizer, ultrasonic sensor, GPS receiver	Helps users reach destination via voice commands	The range of obstacle detection is limited.
[64]	Stick with GSM to detect location	Arduino NANO, LED, Ultrasonic sensors	An effective object detection	Battery life and durability
[65]	Automated navigation system	Raspberry Pi, camera, RFID, ultrasonic sensors	Dual feedback for secure navigation	Distance of objects not calculated
[66]	Smart solution with microcontrollers	Microcontrollers, cameras, accelerometers	Warns users via sound buzzers	Distance of objects not calculated
[67]	Smart walking stick and Smart hat	Buzzers, vibration motors, GPS, ultrasonic sensor	Cost effective system	No object detection

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
