# Peer review of "Assistive Systems for Visually Impaired Persons: Challenges and Opportunities for Navigation Assistance"

_sensors, 2024, doi:10.3390/s24113572_

Round 1

Reviewer 1 Report

Comments and Suggestions for Authors

The paper gives a comprehensive review of existing assistive technology for blind people. Some of the claims need to be documented: Line 465: "Few navigational devices that were created over time to assist the blind and visually impaired are currently in use", " The majority of the systems are too big for most people to carry, hence they are not even seen as navigational aids at all by the user.", "A lot of the current systems are bulky or cumbersome."

Some claims are not documented and not logical: "The reason blind and visually impaired persons use such devices so frequently is due in part to their high cost and the length of time it takes for the user to become acclimated to them.", "Thanks to advancements in computer vision and deep learning technologies, these devices’ features—weight, performance, accuracy, speed, efficiency, and cost—have increased, making them more accessible to those with visual impairments."

Some abbreviations are not sufficiently explained (ROC analysis, VGG16 model, ...). Line 339 - end of the sentence missing. 

Author Response

Comment 1: The paper gives a comprehensive review of existing assistive technology for blind people. Some of the claims need to be documented: Line 465: "Few navigational devices that were created over time to assist the blind and visually impaired are currently in use", " The majority of the systems are too big for most people to carry, hence they are not even seen as navigational aids at all by the user.", "A lot of the current systems are bulky or cumbersome."

Some claims are not documented and not logical: "The reason blind and visually impaired persons use such devices so frequently is due in part to their high cost and the length of time it takes for the user to become acclimated to them.", "Thanks to advancements in computer vision and deep learning technologies, these devices’ features—weight, performance, accuracy, speed, efficiency, and cost—have increased, making them more accessible to those with visual impairments."

Response 1: The section has been written again for clarity purpose and to avoid any misleading information. Please see changes at Page 12, line 455 – 474 in the revised manuscript.

Comment 2: Some abbreviations are not sufficiently explained (ROC analysis, VGG16 model, ...). Line 339 - end of the sentence missing. 

Response 2: We have explained ROC analysis, while VGG 16 is a standard and well-known deep convolutional neural network. Please see the changes on Page 9, line 333 in the revised manuscript.

Reviewer 2 Report

Comments and Suggestions for Authors

Thank you for the opportunity to review this paper . While there is a broad range of information included in the paper, it needs quite a bit of editing Please make sure that you are using the terms visually impaired and blind correctly For example pg 1 line 16 you state "there is an urgent need for efficient assistance for blind... " Do you really mean blind or do you mean visually impaired or both. This is a common occurrence throughout the paper, please check it  

Comments on the Quality of English Language

See attached file  

Author Response

Comment 1: Thank you for the opportunity to review this paper. While there is a broad range of information included in the paper, it needs quite a bit of editing Please make sure that you are using the terms visually impaired and blind correctly for example pg 1 line 16 you state, "there is an urgent need for efficient assistance for blind... " Do you really mean blind or do you mean visually impaired or both. This is a common occurrence throughout the paper, please check it.  

Response 1: We have made correction of this. “Blind” has been changed to “visually impaired” throughout the paper.

Attached PDF Document

Comment 2: In Section 1, line 20 – 23 was highlighted for errors.

Response 2: This was written again properly for clarity. Please see changes in line 23 -24.

Comment 3: In Section 1, line 33. A question was asked “So What?”

Response: The last three words of the sentence was removed. Please see changes in line 30.

Comment 4: In page 2, line 41. Reviewer suggested some words. “Of travel assistive technologies”

Response 4: This has been added to the sentences.   Please see changes in line 38.

Comment 5: In page 6, Reviewer highlighted some inconsistency in the following subheadings “Braille signs, Smart cane, Images into sound, Blind Audio guidance, Voice and Vibration”.

Response 5: These subheadings have all be written again for clarity and consistency.

Comment 6: In Discussions section on page 13, reviewer highlighted some issues.

Response 6: This section has been written again for clarity.

Comment 7: In Conclusions on page 14, reviewer highlighted made some suggestions.

Response 7: These suggestions has been implemented which made us re-write the conclusion.

Reviewer 3 Report

Comments and Suggestions for Authors

This article aims to give a review/state of the art on navigation systems for Visually Impaired people. But the review is essentially on Assistive Systems for Visually Impaired people that can be used as navigational aid. So, I propose to change the title of the Article to "Assistive Systems for Visually Impaired Persons: Challenges and Opportunities for Navigation assistance".

Section 2: It should be: "Assistive Technologies for Visually Impaired". I would suggest to include a Figure 1, with the components that you present in this article: sensors, algorithms feedback mechanisms ...

Paragraph 2.2.4. Haptic Feedback: What kind of haptic feedback is used in the systems you describe (pressure, touch, specific movement or else). Please specify.

Paragraphs 2.3, 2.4 and 2.5 could be merged in "Communication Networks".

Section 3. would be clearer if the title is: "Assistive Tools for Visually Impaired". Table 1 is not necessary (comes as redundant) as it is a summary of all the detailed paragraphs, but they can be added in Fig.1.

Section 4. would be more precise if the title is: "Assistive Systems useful for Navigation of Visually Impaired".

Section 4.1 would more precise if the title is: “Artificial Intelligence-based Object Detection Systems for Visually Impaired”. And the title of Table 2: “Summary of Artificial Intelligence- based Obstacle detection for navigation systems for visually impaired”.

Table 3. presents IoT-based navigation systems with ultrasonic sensors (they all use these sensors). Maybe authors could write a sentence on the safety of using these active sensors.

The sections 5 and 6 are very good. Thanks for this valuable research.

Comments on the Quality of English Language

English is fine

Author Response

Comment 1: This article aims to give a review/state of the art on navigation systems for Visually Impaired people. But the review is essentially on Assistive Systems for Visually Impaired people that can be used as navigational aid. So, I propose to change the title of the Article to "Assistive Systems for Visually Impaired Persons: Challenges and Opportunities for Navigation assistance".

Response 1: Title has been changed as suggested by the reviewer. New title “Assistive Systems for Visually Impaired Persons: Challenges and Opportunities for Navigation Assistance”.

Comment 2: In Section 2: It should be: "Assistive Technologies for Visually Impaired". I would suggest including a Figure 1, with the components that you present in this article: sensors, algorithms feedback mechanisms ...

Response 2: Section 2 title has also been changes as suggested by reviewer.  New section heading “Assistive Technologies for Visually Impaired Persons: Challenges and Opportunities for Navigation Assistance”. Please see changes at Page 3, line 126 in the revised manuscript.

Figure 1 has also been created for components. Page 6

Comment 3: Paragraph 2.2.4. Haptic Feedback: What kind of haptic feedback is used in the systems you describe (pressure, touch, specific movement or else). Please specify.

Response 3: We have provided the detail of the haptic feedback which is “vibration”. Page 5, line 207.

Comment 4: Paragraphs 2.4 and 2.5 could be merged in "Communication Networks".

Response 4: Thanks, and we have been merged paragraphs 2.4 and 2,5 and the new suggested title for section has been implemented "Communication Networks". Page 5, line 222

Comment 5: Section 3. would be clearer if the title is: "Assistive Tools for Visually Impaired". Table 1 is not necessary (comes as redundant) as it is a summary of all the detailed paragraphs, but they can be added in Fig.1.

Response 5: The table has been removed as suggested by reviewer and the content has been added to Fig.1". Page 6

Comment 6: Section 4. would be more precise if the title is: "Assistive Systems useful for Navigation of Visually Impaired".

Response 6: The suggested title for this section has been implemented "Assistive Systems useful for Navigation of Visually Impaired". Page 7, line 273

Comment 7: Section 4.1 would more precise if the title is: “Artificial Intelligence-based Object Detection Systems for Visually Impaired”. And the title of Table 2: “Summary of Artificial Intelligence- based Obstacle detection navigation systems for visually impaired”.

Response 8: The suggested titles for this section have been implemented. “Artificial Intelligence-based Object Detection Systems for Visually Impaired”. Page 7, line 280

And the updated title of Table 2: “Summary of Artificial Intelligence- based Obstacle detection navigation systems for visually impaired”. Page 8

Comment 9: Table 3. presents IoT-based navigation systems with ultrasonic sensors (they all use these sensors). Maybe authors could write a sentence on the safety of using these active sensors.

Response 9: A sentence with a reference has been provided to this at the last paragraph of this section. Page 12, line 450

Comment 10: The sections 5 and 6 are very good. Thanks for this valuable research.

Response 10: Thanks for the detailed feedback to improve the quality of the paper.

Round 2

Reviewer 2 Report

Comments and Suggestions for Authors

Thank you for the updates you made to your article. I have a few more comments that I hope will be helpful 

Pg 1 line 11 "At least 2.2 billion people suffer ( not suffers).

pg 12 line 458-460 " It has been reported that visually impaired people find most systems irritating and depressing... " DO you have a citation for this statement - How do you know this is the case? 

pg 12 line 466. Instead of saying " the significance of AT... This editor suggests saying 'The availability... 

On line 467 at the end of the sentence you state "capacities, security and self assurance " I am not sure what you mean by self assurance 

pg 12 469 The line that begins Deep learning methods ... can be integrated hence increasing efficiency and USE - how do you know use will be increased could you state instead " increasing efficiency and promise of these technologies?

On pg 13 line 478-9 you state It is recommended that the navigation solution concentrate on communicating certain environmental information to be effective certain environmental information such aas what ? can you give an example? 

pg 13 482-487 " however depending on the circumstances, each of these modes will occasionally become significant - I am not sure what you are trying to say here can you reword this section up through line 487 to clarify what you are trying to say here 

pg 13 line 490 What does " both of these" refer to ? 

pg 13 491 -493 One of the factors .. is that customers should not have any trouble picking up ( are you trying to say it shouldnt be too heavy or too complicated to learn - please clarify 

pg 13 494-495 "One of the key" maybe change to Another key element influencing usefullness and user adoption of these gadgets ( consider using the word devices) is portability  DO you have a citation for this statement How do you know portability influences adoption and use ? 

and finally

The last sentence The review .... is having a significant influence on both the reasearch community and society at large ? How do you know this  

        Comments on the Quality of English Language

Author Response

Comment 1: Pg 1 line 11 "At least 2.2 billion people suffer (not suffers).

Response 1: This has been corrected to “suffer”. Please see changes in page 1, line 11.

Comment 2: pg 12 line 458-460 " It has been reported that visually impaired people find most systems irritating and depressing... " DO you have a citation for this statement - How do you know this is the case? 

 Response 2: The statement has been removed to avoid confusion and improve the readability of the work. Please see changes in page 12.

Comment 3: pg 12 line 466. Instead of saying " the significance of AT... This editor suggests saying 'The availability... 

Response 3: This has been corrected to “The availability of”. Please see changes om page 12 line 466.

Comment 4: On line 467 at the end of the sentence you state "capacities, security and self-assurance " I am not sure what you mean by self-assurance 

Response 4: “Self-assurance” has been removed from the sentence for clarity. Please see changes in line 467.

Comment 5: pg 12 469 The line that begins Deep learning methods ... can be integrated hence increasing efficiency and USE - how do you know use will be increased could you state instead " increasing efficiency and promise of these technologies?

Response 5: This has been corrected to “potentials” for better flow of the sentence. Please see changes in line 471.

Comment 6: On pg 13 line 478-9 you state It is recommended that the navigation solution concentrate on communicating certain environmental information to be effective certain environmental information such as what? can you give an example? 

Response 6: The statement has been removed as it was just an extension of the previous statement. Please see changes in page 13.

Comment 7: pg 13 482-487 " however depending on the circumstances, each of these modes will occasionally become significant - I am not sure what you are trying to say here can you reword this section up through line 487 to clarify what you are trying to say here. 

Response 7: The paragraph has been written again for clarity. Please see changes from line 480 – 484.

Comment 8: pg 13 line 490 What does " both of these" refer to? 

Response 8: The statement has been adjusted for better understanding. Please see changes in line 487.

Comment 9: pg 13 491 -493 One of the factors .. is that customers should not have any trouble picking up ( are you trying to say it shouldnt be too heavy or too complicated to learn - please clarify 

 Response 9: The sentence has been written again for clarity and better understanding. Please see changes from line 489 – 490.

Comment 10: pg 13 494-495 "One of the key" maybe change to Another key element influencing usefulness and user adoption of these gadgets (consider using the word devices) is portability  DO you have a citation for this statement How do you know portability influences adoption and use ? 

Response 10: The words have been adjusted and a reference have been provided. Please see changes between line 491 – 494.

Comment 11: The last sentence The review .... is having a significant influence on both the research community and society at large? How do you know this  

Response 11: It states the review demonstrates this “field of study”. It’s not talking about this review paper only, but the field of assistive technologies for the visually impaired people at large. Please see the full sentence in the last paragraph.

Reviewer 3 Report

Comments and Suggestions for Authors

This article aims to give a review/state of the art on assistive systems for Visually Impaired people that could be used while they are following a trajectory. The corrections requested previously are applied by authors. There are still some sentences to correct:

Page 3, Line 119: Analysis of Artificial Intelligence- based object detection systems...

Line 122: Sention 2 discuss assistive tools, ...

Line 124: assistive systems based on their technologies.

2.1. Visually chellenged people frequently employ assistive tools that use depth cameras...

Page 5, Line 224: What do you mean by dependable ?

Page 6, Fig. 1, I suggest "Tools" instead of Names.

Table 1. obstacle detection systems for visually...

Comments on the Quality of English Language

English language is fine

Author Response

Comment 1: Page 3, Line 119: Analysis of Artificial Intelligence- based object detection systems...

Response 1: This has been corrected and now written as “Artificial Intelligence- based object detection systems”. Please see changes on page 3, line 119.

Comment 2: Line 122: Section 2 discuss assistive tools, ...

Response 2: This has been corrected and now written as “assistive tools”. Please see changes in line 122.

Comment 3: Line 124: assistive systems based on their technologies.

Response 3: This has been corrected and now written as “assistive systems based on their technologies”. Please see changes in line 124.

Comment 4: 2.1. Visually challenged people frequently employ assistive tools that use depth cameras...

Response 4: This has been corrected and now written as “Visually challenged people frequently employ assistive tools that use depth cameras”. Please see changes in section 2.1.

Comment 5: Page 5, Line 224: What do you mean by dependable?

Response 5: The word “dependable “has been removed for clarity and better understanding. Please see changes in page 5, line 224.

Comment 6: Page 6, Fig. 1, I suggest "Tools" instead of Names.

Response 6: This has been corrected and now written as “Tools”. Please see changes in page 6, fig 1.

Comment 7: Table 1. obstacle detection systems for visually...

Response 7: This has been corrected and now written as “obstacle detection systems for visually”. Please see changes in page 8.